# The Ligurian Experience in the Management of Lung Cancer: Organizational Models and New Perspectives

**DOI:** 10.3390/healthcare12242556

**Published:** 2024-12-18

**Authors:** Daniela Amicizia, Francesca Marchini, Paolo Pronzato, Gabriella Paoli, Carlo Genova, Silvia Allegretti, Filippo Ansaldi

**Affiliations:** 1Ligurian Regional Health Service, Piazza della Vittoria 15, 16121 Genoa, Italy; daniela.amicizia@alisa.liguria.it (D.A.); gabriella.paoli@alisa.liguria.it (G.P.); filippo.ansaldi@alisa.liguria.it (F.A.); 2Department of Health Sciences, University of Genoa, 16132 Genoa, Italy; silvia.allegretti@alisa.liguria.it; 3IRCCS San Martino Hospital, 16132 Genoa, Italy; paolo.pronzato@hsanmartino.it (P.P.); carlo.genova@hsanmartino.it (C.G.); 4Department of Internal Medicine and Medical Specialties, University of Genoa, 16132 Genoa, Italy

**Keywords:** lung cancer, lung cancer treatment, tumor care, hub and spokes

## Abstract

Background: Lung cancer is an oncological threat worldwide, including in Italy. New organizational approaches based on a network of cancer centers and multidisciplinary and technological innovation are required. The experience in the Liguria region, in northwestern Italy, in the management of lung cancer is presented with a focus on the organizational model. Methods: A retrospective observational analysis was conducted for the period from January 2019 to December 2023 using administrative regional data. Results: Of the total surgery treatments in Liguria, most were carried out at the IRCCS San Martino (about 47%), which is the hub’s center. Most cases involved males aged ≥65 years (*p* < 0.001). Passive mobility showed a decrease in recent years. Considering the type of access to clinical structures, almost all that were finalized to receive chemotherapy were from the day hospital regimen (99%). Conclusion: A comprehensive approach must be carried out for cancer patients to maintain high levels of care quality. In this challenging context, the Liguria region has implemented new organizational approaches based on the networking of cancer centers and multidisciplinary and technological innovation.

## 1. Introduction

Lung cancer is an oncological threat worldwide, including in Italy [1,2,3,4]. In 2023, approximately 44,000 new cases of lung cancer were diagnosed in Italy with men being disproportionately affected (30,000 cases) compared to women (14,000 cases). The prognosis for lung cancer patients is variable with a 5-year survival rate of only 16% for men and 23% for women [4]. In 2022 alone, lung cancer was responsible for about 35,700 deaths in the country [5].

Over the past two decades, the treatment of lung cancer has completely changed, moving from surgery, radiotherapy, and chemotherapy treatments to more widely available new therapeutic strategies based on effectively targeted agents (for oncogene-driven cases) and immune checkpoint inhibitors [6].

Mainly due to the introduction of these new antitumor drugs, the prognosis of lung cancer patients has improved toward a prolongation of survival for patients suffering from advanced disease and a higher cure rate for patients with non-metastatic disease fit for a multimodality approach that includes local therapies and newer drugs [7].

Most of these successes have been achieved by the implementation of diagnostic and staging processes that are able to detect the extension of disease and the presence of biomarkers as drivers of the clinical decisional process. In particular, access to next-generation sequencing (NGS) and other modalities of oncogene alteration detection has been shown to be fundamental in choosing among the many new targeted drugs [8].

The sustainability of public universalistic health systems, including the Italian National Health System (INHS), is being challenged by many factors [9], such as the high impact of lung tumors on healthcare services and the high demand for diagnostics and treatments.

Regarding tumor care, the crude incidence increase (due to the aging of the population), the shortage of specialized personnel, the high cost of innovative technologies, and the chronicization of advanced disease represent major issues [10].

The incidence of lung cancer is projected to keep rising in most countries until 2035, establishing lung cancer as a significant global public health issue. This evolving epidemiological landscape underscores the necessity of reallocating resources and enhancing lung cancer control strategies to mitigate its future impact [11].

It is crucial to assess existing organizational models with a focus on continuous improvement, particularly to address the growing number of cases and to optimize available resources. Indeed, in order to maintain high levels of care quality in this challenging context, the INHS and its branches have implemented new organizational approaches based on the networking of cancer centers and multidisciplinary and technological innovation.

In the present study, the experience in the Liguria region, northwest Italy, in the management of lung cancer is presented with a focus on the organizational model and the future challenges in this field.

## 2. Materials and Methods

### 2.1. Demographic Characteristics

Liguria is an administrative region in the northwest of Italy with a population of approximately 1,509,227 million inhabitants; it is the oldest Italian region with an aging index of 270.8% [12].

### 2.2. Setting and Organizational Models

The regional healthcare system is organized into five Local Health Units (LHUs) covering hospital and territorial healthcare; furthermore, four other hospitals are also operating. Specifically, the IRCCS San Martino Hospital is recognized by the Italian Minister of Health as a “comprehensive cancer center” for adults. Furthermore, the Regional Health Agency of Liguria (A.Li.Sa.) performs healthcare management and planning tasks.

Regarding surgical lung cancer units, a hub-and-spoke model is applied, including a thoracic surgery unit located at the IRCCS San Martino Hospital, which is the hub’s center.

The units for the specialized care of chronic cancer patients are capillary with five radiotherapy units and nine medical oncology units, which are authorized for the administration of antitumor agents (Appendix A).

The Liguria region is among the first Italian regions in the past decade to implement an oncological network in order to ensure quality care and guarantee access for all cancer patients. This model is based on a Clinical Governance Committee (in charge of the production of Journey Documents) and the deep reorganization of hospital assistance based on a multidisciplinary approach. Indeed, local Disease Management Teams (DMTs) include specialists involved in lung cancer diagnosis and treatment, such as medical oncologists, radiation oncologists, pulmonologists, radiologists, pathologists, molecular biologists, nuclear medicine specialists, psychologists, palliative care specialists, and case managers, who regularly meet on a weekly basis and discuss the most appropriate pathway for each patient. Notably, the discussed management of patients who are eligible for treatments includes cyto-histological diagnosis, molecular characterization (e.g., next-generation sequencing), and treatment, which involves surgery or radiotherapy for localized non-small cell lung cancer (NSCLC); chemo-radiation for unresectable, locally advanced NSCLC or for limited small cell lung cancer (SCLC); systemic treatment, which includes immunotherapy-based regimens for metastatic, non-oncogene-addicted NSCLC and extensive SCLC; or targeted therapies for oncogene-addicted NSCLC. Such treatments are managed according to current national guidelines [13]. Additionally, DMTs are designed to periodically assess the critical issues of the lung cancer pathway, especially for particular patient subgroups, with the aim of proposing possible solutions in order to constantly improve the access to diagnosis and treatment throughout the Liguria region [14,15,16].

Furthermore, the DMTs are designed to manage other aspects of cancer care, such as simultaneous care, psychological support, nutrition, palliation, fertility preservation, etc. As regards simultaneous care, it guarantees the most appropriate care setting during the different trajectories of the disease, reduces care costs and the use of chemotherapy in the last 30 days of life, improves communication between the oncologist and the patient and alleviates the anxiety and depression of caregivers. They are the true connection and integration between hospital and territorial services dedicated to the advanced stage oncological patient and it is important to anticipate their needs.

Multidisciplinary groups are identified within the network in order to ensure the best patient pathway by defining the role and responsibilities for each involved professional figure, including the case manager. The “multidisciplinary” and “multiprofessional” group is made up by specialized physicians, nurses and other healthcare workers who establish the most appropriate diagnosis and treatment pathway after an overall assessment of the patient.

Furthermore, the need to characterize lung cancer in term of genetic alterations that may act as oncogene drivers, and thus be exploitable for targeted therapies, has led to a series of initiatives aimed at concentrating technologies at San Martino Hospital. Governance is provided by a Regional Molecular Tumor Board, while the management of therapies remains with each oncology unit in the region. 

Lastly, it is important to highlight the significant work carried out by volunteer associations that support oncology patients and their caregivers throughout their journey [17].

### 2.3. Study Design

A retrospective observational analysis was conducted over the period from 1 January 2019 to 31 December 2023.

Data were collected from the administrative healthcare data “Data warehouse” service, which includes Hospital Discharge Records (HDRs) as well as records of patient visits to the medical oncology units for Day-Hospital care dedicated to outpatients.

Oncologic patients were included on the basis of the International Classification of Diseases 9th Revision, Clinical Modification (ICD-9-CM) diagnosis and procedure codes selecting patients who were admitted with a diagnosis of Primary Lung Cancer (ICD-9-CM codes: 162.x; V10.11) and surgery procedures (ICD-9-CM codes: 32.x).

Resection, lobectomy, or pneumonectomy (ICD-9-CM 32.X) were considered.

### 2.4. Statistical Analysis

Given the observational nature of the study, a descriptive analysis was performed in order to characterize the data set, its variability, and its distribution.

The data extracted from the regional data warehouse were exported to Microsoft Excel 2021 and subsequently analyzed using JMP^®^ Pro V17 Statistical Software [18].

The analysis of variance and parameter estimates was carried out, and the effect test was applied.

Categorical variables were summarized as frequencies and percentages; comparisons were made using chi-square tests (χ^2^).

## 3. Results

Between 2019 and 2023, a total of 1744 surgical interventions aimed at treating pulmonary cancer were performed in the Liguria region (Table 1).

Surgical interventions, broken down by area of residence (LHU) and year, are shown in Table 1. Regarding the amounts of surgical interventions, no statistical differences were observed across the years (*p* > 0.001).

As expected, the majority of curative surgeries have been carried out at the hub center IRCCS San Martino (about 65.3%), which is followed by the other centers such as LHU3 and LHU5.

Regarding the indicator “mortality within 30 days”, an adjusted value of 0.59 (CI 95% 0–2.91) was registered in 2023.

Table 2 reports surgical interventions performed during the study period by gender and age. The majority of cases were concentrated in the male group aged ≥65 years (*p* < 0.001).

As the Liguria region is characterized by an aging population, this demographic factor favors an increased risk of developing lung cancer, which is a disease primarily affecting older populations.

Furthermore, during the reference period, 590 Ligurian patients underwent surgical intervention in other Italian regions, resulting in an overall passive health migration rate of 33.7%; this rate decreased to 25.3% in 2023.

In Figure 1, production (green), passive mobility (orange) and active mobility (blue) are illustrated.

An increase in production was observed in the study period, which was accompanied by a reduction in passive mobility. Regarding access to clinical facilities, nearly all visits for systemic treatments were conducted under a “day hospital” regimen (99%) with no instances of inpatient day service recorded.

In terms of hospitalizations, a significant difference was detected between subjects 18–64 years and those aged ≥65 years (*p* < 0.001) across all considered periods (Table 3A,B).

## 4. Discussion

Liguria is among the first Italian regions that implemented different organizational models: for surgery treatment, a hub-and-spoke approach has been adopted since 2019, while a proximity delivery model has been applied for systemic therapies such as chemotherapy, immunotherapy or targeted therapy [19]. This model leverages multiple radiotherapy and medical oncology units, which are aimed at providing treatment for chronic cancer patients. The organizational structure aims at enhancing clinical outcomes and improving the quality of services for pulmonary cancer patients.

A significant disparity is observed between the surgical and medical components of the therapeutic journey. While surgical procedures are performed at a limited number of sites within the region—and for some patients, even outside the region—the majority of patients receive systemic antineoplastic treatments at hospitals relatively close to home. This is logical, given that each treatment line may span several months or even years. Specifically, regarding antineoplastic treatment, all operating units provide cycles of therapy under a proximity delivery model, ensuring accessible care within the regional healthcare system and facilitating treatment for patients.

Our findings align with the official regional programmatic documents, which emphasize the centralization of complex cases in highly specialized centers, adherence to official Italian standards, and the establishment of an extensive network for cancer treatments [20].

Non-small cell lung cancer (NSCLC), the most frequent form of lung cancer, is typically treated with curative surgery when the tumor is radically operable, and resection can eventually be followed by systemic treatments such as chemotherapy and immunotherapy to reduce the risk of recurrence [21]. In contrast, locally advanced, inoperable NSCLC can be managed with a combination of chemotherapy and radiotherapy, which can be potentially followed by maintenance immunotherapy. The treatment of metastatic disease primarily involves systemic therapies, including targeted therapies, immune checkpoint inhibitors, and/or chemotherapy, tailored to the tumor’s clinical, pathological, and molecular characteristics.

Notably, the introduction of targeted therapies and immune checkpoint inhibitors into the available therapeutic options has significantly improved patient outcomes across all the disease stages. However, this advancement has also led to increased costs and greater complexity in oncological care [21,22,23,24,25,26].

Nonetheless, lung cancer mortality remains still high in Italy [5]; in 2022, an estimated 35,700 deaths from lung cancer (men = 23,600; women = 12,100) were registered. This high mortality is at least partially linked to the frequent diagnosis of lung cancer at advanced stages, which is primarily due to the absence of systematic screening programs.; furthermore, diagnoses carried out after the onset of severe symptoms is typically associated with a significantly worse prognosis [14].

In this context, precision medicine approaches introduce an additional layer of complexity to the therapeutic landscape of lung cancer, as multiple genic alterations can be analyzed simultaneously through NGS. This allows the identification of oncogenic drivers with potential therapeutic implications [27]. Specifically, in Liguria, more than 400 tests were requested in 2023 for lung cancer profiling with the activity showing notable growth in the recent period. In more than 50% of cases, a targetable mutation for a targeted therapy was identified.

In 2019, the regional clinical therapeutic care pathways of patients suffering from cancers were defined by means of a regional resolution [28]; they are tools used to guide evidence-based healthcare, allowing translating clinical practice guideline recommendations into clinical processes of care within the unique culture and environment of a healthcare institution [29]. They provide a valuable approach for appraising and synthesizing evidence to support policy development and quality improvement. In this context, tumor boards and disease management teams play a key role in treatment planning with clinicians from various specialties collaboratively discussing the clinical status and treatment options for each patient. These boards are essential for generating coordinated treatment plans and ensuring timely access to care [30]; they help improve adherence to guidelines and mitigate potential bias from the treating or diagnosing physician. Furthermore, tumor boards function not only as mechanisms for improving patient care but also as important educational forums, offering significant learning opportunities for postgraduate trainees at university institutions [31].

Despite the good performance of lung cancer treatment in Liguria, passive mobility, i.e., patients residing in Liguria seeking treatment outside the region, remains. However, it is relevant to note that this phenomenon is a decreasing trend. This is very positive for economic and social implications.

Today, advanced cancer patients are often considered chronic patients, as the time to terminality and death has been significantly prolonged. Pharmacological antitumor therapy typically involves a series of treatment lines, some of which are based on oral medications, and it is guided by a multidisciplinary team and supported by qualified oncological surveillance. Alongside treatment, supportive and palliation are also essential. The patient journey often spans various care setting (hospital, clinics, home care, rehabilitation centers and hospice).

In this context, it is crucial to address the access challenges faced by patients and caregivers. For antitumor medical treatments, high-quality care is ensured by the proximity of medical oncology units.

Interestingly, our research crosses the COVID age: in 2020, a decline in surgical intervention volumes was observed, but medical treatment volumes remained stable [32]. The decrease in surgeries during the pandemic period was likely due to the limitations in diagnostic procedures rather than limitations in the availability of surgery, which was still ensured by the Regional Healthcare Service [RHS]). Other studies did not find similar results for surgery [33]. However, after the initial months of the pandemic, the healthcare system was gradually reorganized in order to ensure sufficient levels of care for lung cancer [34].

In conclusion, the regional organization appears well equipped to manage the evolving needs of lung cancer patients, providing access to vital therapies through the medical oncology units at all hospitals. However, further adjustments are necessary, particularly given the rapid expansion of (neo)adjuvant therapies for early and locally advanced lung cancer. These approaches require prompt molecular characterization and perioperative medical treatments under the guidance of an experienced team. Additionally, there is a growing need for access to personalized medications in many cases, which is closely tied to the ability to reach specialized centers.

Furthermore, lung cancer screening will play a crucial role in the future by detecting lung cancer before symptoms appear through imaging tests, when it is still potentially curable [35]. 

Finally, healthcare planning research may be essential in redefining processes and organizational structures; additionally, the exploration of the vast amount of recorded data will facilitate the generation of real-world evidence.

The hub-and-spoke model, along with updated diagnostic–therapeutic pathways and harmonized healthcare delivery across the region, could significantly improve the patient services, particularly in terms of outcomes [36]. While pharmaceutical and other modern technologies continue to advance, the health systems have adopted contemporary cancer care models, including a multidisciplinary approach, oncological networks, and the implementation of supportive care such as nutrition, psycho-oncology, and the fertility preservation. Furthermore, early and simultaneous palliative care is being integrated.

Considering the profound impact of lung cancer on patients’ and their families’ physical, emotional, and social well-being, it is critical to address various aspects of care, including support for caregivers [37,38].

The ongoing challenge for the Regional Health Service (RHS) is to consistently fulfill its responsibility of safeguarding public health, ensuring the provision of essential healthcare services while simultaneously maintaining the sustainability of the services offered to citizens.

The limitations of this study include the lack of investigation into the specific therapies used in lung cancer treatments and the absence of granular data on patient pathways, as these were not available due to restrictions imposed by the regional dataset. Additionally, performance indicators related to the diagnostic and therapeutic pathways, which could have provided a more comprehensive evaluation of the care quality and treatment effectiveness, were not reported. These aspects will be addressed in future analysis. On the other hand, the study’s strengths lie in the use of cutting-edge surgical techniques at the hub center IRCCS San Martino, which serves as a regional reference for lung cancer treatment. Furthermore, the study was based on “real world” data, drawn from actual clinical cases, allowing for a practical assessment of the organizational model’s impact. The integration of technological innovation and a multidisciplinary approach has provided valuable insights into how new organizational strategies can enhance lung cancer management in Liguria.

## 5. Conclusions

Considering the current context of limited resources, scientific advancements and growing demand for care, there is an urgent need to identify and adopt clinical, organizational and management processes aimed at achieving ever greater efficiency, appropriateness and equity of care. A more integrated approach to healthcare data, combined with organizational and managerial information, can result in increased quality of care in all its dimensions. Healthcare planning plays a crucial role in collecting and analyzing data in order to generate evidence that can support the development of increasingly cutting-edge care models.

## Figures and Tables

**Figure 1 healthcare-12-02556-f001:**
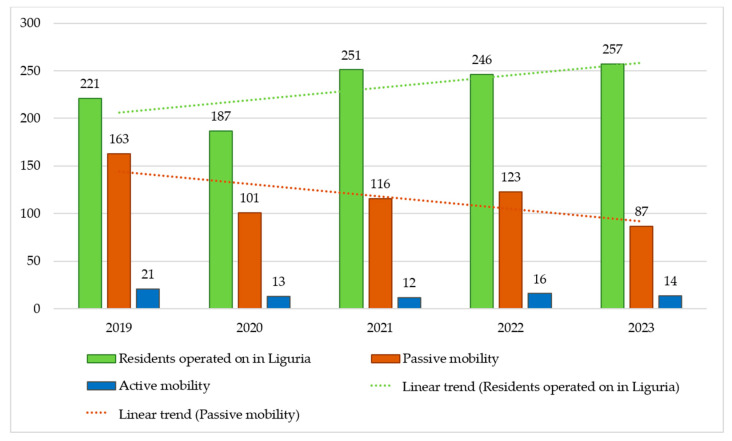
Production of surgical interventions, active and passive mobility by year.

**Table 1 healthcare-12-02556-t001:** Surgical interventions and their distribution by area of residence (LHU) from 2019 to 2023.

	Resident Patients							
Year	Providing Health Company	LHU1	LHU2	LHU3	LHU4	LHU5	Not Defined	Not Ligurian
2019	LHU1	0	0	0	0	0	0	0
	LHU2	*	18	0	0	0	0	*
	LHU3	*	0	58	*	*	0	*
	LHU4	0	0	0	0	0	0	0
	LHU5	0	0	*	0	25	0	0
	IRCCS San Martino H	*	*	86	13	5	*	11
	Other Hospitals	0	0	0	0	0	0	*
	Out of Region	23	46	43	32	19	15	0
2020	LHU1	0	0	0	0	0	0	0
	LHU2	*	*	0	0	0	0	*
	LHU3	*	0	32	*	0	0	0
	LHU4	0	0	0	0	0	0	0
	LHU5	0	0	0	0	18	*	*
	IRCCS San Martino H	11	21	79	10	5	*	7
	Other Hospitals	0	0	0	0	0	0	*
	Out of Region	15	27	28	13	18	6	0
2021	LHU1	0	0	0	0	0	0	0
	LHU2	0	0	0	0	0	0	0
	LHU3	*	0	37	0	*	0	*
	LHU4	0	0	0	0	0	0	0
	LHU5	0	0	0	0	26	0	*
	IRCCS San Martino H	21	35	104	16	8	0	9
	Other Hospitals	0	0	0	0	0	0	*
	Out of Region	0	0	0	0	0	0	0
2022	LHU1	0	0	0	0	0	0	0
	LHU2	0	0	0	0	0	0	0
	LHU3	0	0	45	*	0	0	0
	LHU4	0	0	0	0	0	0	0
	LHU5	0	0	0	0	28	0	*
	IRCCS San Martino H	33	32	90	11	5	*	13
	Other Hospitals	0	0	*	0	0	0	0
	Out of Region	11	33	34	17	28	8	0
2023	LHU1	0	0	0	0	0	0	0
	LHU2	0	0	0	0	0	0	0
	LHU3	*	0	68	*	0	*	0
	LHU4	0	0	0	0	0	0	0
	LHU5	0	0	0	0	25	0	*
	IRCCS San Martino H	36	31	81	14	0	*	9
	Other Hospitals	0	0	0	0	0	*	0
	Out of Region	13	18	19	14	23	6	0

(*) value < 5. It is not possible to provide detailed data due to privacy restrictions.

**Table 2 healthcare-12-02556-t002:** Surgical interventions broken down by gender and age.

		MALES				FEMALES			
Year	Providing Health Company	18–44 Years	45–64 Years	65–74 Years	≥75 Years	18–44 Years	45–64 Years	65–74 Years	≥75 Years
2019	LHU2	0	6	*	*	0	*	*	*
	LHU3	0	5	15	21	0	6	7	14
	LHU5	0	7	5	6	0	*	*	5
	IRCCS San Martino H	0	17	29	19	*	15	24	16
2020	LHU2	0	0	*	*	0	*	*	0
	LHU3	0	*	11	11	0	*	7	*
	LHU5	*	*	*	*	*	*	*	*
	IRCCS San Martino H	*	21	40	21	*	18	19	9
2021	LHU2	0	0	0	0	0	0	0	0
	LHU3	0	*	10	11	*	*	12	*
	LHU5	0	5	9	6	0	*	*	*
	IRCCS San Martino H	*	22	41	35	*	34	35	23
2022	LHU2	0	0	0	0	0	0	0	0
	LHU3	0	8	10	11	0	6	7	*
	LHU5	0	*	7	8	0	*	7	*
	IRCCS San Martino H	*	14	43	37	*	21	24	29
2023	LHU2	0	0	0	0	0	0	0	0
	LHU3	*	6	15	18	0	8	15	8
	LHU5	0	0	*	12	0	*	6	*
	IRCCS San Martino H	*	14	43	37	*	21	24	29

(*) value < 5. It is not possible to provide detailed data due to privacy restrictions.

**Table 3 healthcare-12-02556-t003:** Hospitalizations, days of hospitalization and type of regimen broken down by year and age class (*p* value < 0.001).

**(A)—Age Class Under 65**						
	**18–64 years (Age Class)**					
**Year**	**Hospitalizations (n)**	**Ordinary Regimen**	**Hospitalization (days)**	**Surgical Interventions (n)**	**Day Hospital Regimen (n)**	**Day Hospital Regimen (days)**
2019	266	266	3141	64	56	273
2020	242	242	2750	67	49	291
2021	225	225	2435	74	37	173
2022	256	256	2815	77	50	233
2023	223	223	2426	55	53	213
**(B)—Age Class ≥ 65**						
	**≥65** **Years (Age Class)**					
**Year**	**Hospitalizations (n)**	**Ordinary Regimen**	**Hospitalization (days)**	**Surgical Interventions (n)**	**Day Hospital Regimen (n)**	**Day Hospital Regimen (days)**
2019	853	853	10,677	175	152	621
2020	713	713	8962	133	136	577
2021	748	748	8923	188	149	582
2022	853	853	9452	184	170	651
2023	861	861	10,665	215	199	575

## Data Availability

The data are not publicly available due to privacy and ethical restrictions.

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
