# Peer review of "The Ligurian Experience in the Management of Lung Cancer: Organizational Models and New Perspectives"

_healthcare, 2024, doi:10.3390/healthcare12242556_

Round 1

Reviewer 1 Report

Comments and Suggestions for Authors

The authors are presenting the model of medical care in one italian region, Liguria. The main issue was the surgical management but this is only a small part of the patient care. Indeed data are only be given for surgery while no information is provided for the other disciplines. We just have the information on a networking but we have no information on the networkingt function in  daily practice. Do you have a multidisciplinary tumor board? this is just one question.  The paper is too long and in the discussion there is a lot of comments on the current management of lung cancer but not related to the data presented.par

Author Response

Reviewer 1

The authors are presenting the model of medical care in one Italian region, Liguria. The main issue was the surgical management, but this is only a small part of the patient care. Indeed, data are only be given for surgery while no information is provided for the other disciplines. We just have the information on a networking but we have no information on the networking function in daily practice. Do you have a multidisciplinary tumor board? this is just one question.  The paper is too long and in the discussion, there is a lot of comments on the current management of lung cancer but not related to the data presented.par

Reply

We thank the reviewer for his comments. We added the information on the clinical pathway and multidisciplinary tumor board, which is called disease management team (DMT) in our context.

As required the discussion has been shortened (lines 180-188; 244-246).

Reviewer 2 Report

Comments and Suggestions for Authors

The chosen topic of the study is interesting and important. However, the information provided in the manuscript is quite chaotic, it lacks basic information, without which it is impossible to evaluate the lung cancer care model (which is the aim of the study).

1. One table should contain at least the following essential information:

• How many lung cancer cases were there each year in total and how many in each department.

• What stages according to TNM were there cases each year in total and how many in each department.

• What histological forms of lung cancer were there in percentage.

• What percentage of adenocarcinomas were EGFR and ALK mutated.

• What percentage of cancer patients were treated surgically (radical surgery), radical radiotherapy, palliative radiotherapy, chemoradiotherapy, only chemotherapy, biological therapy, immunotherapy in general, and by departments.

• What percentage of subjects underwent testing for EGFR, ALK and other cancer markers, testing for PD-l1.

2. There is no information about the typical path of investigation and care of patients with lung cancer in the region.

3. What proportion of patients were not suitable for any specific lung cancer treatment?

4. What proportion of patients underwent PET-CT, brain MRI, EBUS examination of mediastinal lymph nodes.

5. It is unclear what is meant by "surgical intervention".

Author Response

Reviewer 2

The chosen topic of the study is interesting and important. However, the information provided in the manuscript is quite chaotic, it lacks basic information, without which it is impossible to evaluate the lung cancer care model (which is the aim of the study).

  1. One table should contain at least the following essential information:
  • How many lung cancer cases were there each year in total and how many in each department.
  • What stages according to TNM were there cases each year in total and how many in each department.
  • What histological forms of lung cancer were there in percentage.
  • What percentage of adenocarcinomas were EGFR and ALK mutated.
  • What percentage of cancer patients were treated surgically (radical surgery), radical radiotherapy, palliative radiotherapy, chemoradiotherapy, only chemotherapy, biological therapy, immunotherapy in general, and by departments.
  • What percentage of subjects underwent testing for EGFR, ALK and other cancer markers, testing for PD-l1.

Reply

Thank you for your suggestions, however the study protocol approved by Ethical Committee foresee the use of data from regional database. Therefore, the information about single patient is not registered in the regional data warehouse, the only allowed source that can we use for this exploratory study.

In future research supported by specific protocol and involving healthcare Companies, we will investigate the pathway of oncologic patients. This is a limit of the present study.

  1. There is no information about the typical path of investigation and care of patients with lung cancer in the region.

We agree with reviewer comment, therefore a sentence of clinical pathway has been added.

  1. One table should contain at least the following essential information...

Not feasible

  1. What proportion of patients were not suitable for any specific lung cancer treatment?

Not feasible

  1. What proportion of patients underwent PET-CT, brain MRI, EBUS examination of mediastinal lymph nodes.

Not feasible

  1. It is unclear what is meant by "surgical intervention".

For surgical intervention a decsription of ICD9-CM codes have been reported in methods section.

Specifically, resection, lobectomy, or pneumonectomy (ICD-9-CM 32.X) were considered.

Reviewer 3 Report

Comments and Suggestions for Authors

This paper presents data from a specific geographical region. Some points that need a major revision can be found below:

In the Introduction authors need to provide more references in order to support the rationale of their research.

How were participants recruited. Please describe in more detail in the Methods section.

Authors should add in the discussion information about what happens with the caregivers in this system as well as in other cultural context (for relevant articles on this to use and discuss interventions and proposals for future changes in the healthcare system for this disease: https://doi.org/10.1177/1479972311433577

https://doi.org/10.1002/pon.5271)

Why was JMP software used instead of others? Please justify.

Furthermore, for quality care mood disorders should also be addressed in this group of patients as they are common e.g. depression especially for patients over 65 years of age that is older adults which are common in your sample (doi: 10.1002/pon.4422) apart from the proposed solely technological interventions in medical treatment.

Mainly descriptive statistics are presented. Maybe more complex statistical analyses should be added and included in the hypotheses.

Author Response

Reviewer 3

This paper presents data from a specific geographical region. Some points that need a major revision can be found below:

  1. In the Introduction authors need to provide more references in order to support the rationale of their research.
  2. How were participants recruited. Please describe in more detail in the Methods section.
  3. Authors should add in the discussion information about what happens with the caregivers in this system as well as in other cultural context (for relevant articles on this to use and discuss interventions and proposals for future changes in the healthcare system for this disease: https://doi.org/10.1177/1479972311433577 https://doi.org/10.1002/pon.5271)

Furthermore, for quality care mood disorders should also be addressed in this group of patients as they are common e.g. depression especially for patients over 65 years of age that is older adults which are common in your sample (doi: 10.1002/pon.4422) apart from the proposed solely technological interventions in medical treatment.

  1. Mainly descriptive statistics are presented. Maybe more complex statistical analyses should be added and included in the hypotheses.
  2. Why was JMP software used instead of others? Please justify.

Reply

  1. As required more references have been added in the introduction.
  2. This is an explorative retrospective analysis was conducted using data obtained through the administrative healthcare “data warehouse” regional service of the Hospital Discharge Records (HDRs) and the flow of outpatient visits.
  3. We added a sentence on caregivers and the related two references.
  4. We thank reviewer comment, but as reported in methods section, we preferred to carry out only basic descriptive statistical analysis for this explorative study. Furthermore, aggregated data allow us to perform only a descriptive analysis.
  5. We used JMP as support for statistical analysis as our department has the licence of this software.

Round 2

Reviewer 1 Report

Comments and Suggestions for Authors

Thank you for taking into account my comments

Author Response

The editor reported: "I thank and congratulate the authors for their replies to the reviewers, having addressed what it was possible and having motivated the reasons for what it was not possible to address, clearly reporting the limitations of their study.
I have only one note to add: section 2.4 (Statistical Analyses) seems too poor to me; I think that, though only descriptive analysis has been performed, something more could be said about the descriptive analysis done; moreover, a reference and the version of JMP software used should be added."

Therefore, we added the description of the statistical analyses and the corresponding reference.

Reviewer 2 Report

Comments and Suggestions for Authors

Unfortunately, the manuscript is essentially uncorrected. The fundamental questions I raised have not been answered, in fact, they have been ignored.  

Author Response

(The authors gave the same response as above.)
